# The Addition of Hot Water Extract of Juncao-Substrate *Ganoderma lucidum* Residue to Diets Enhances Growth Performance, Immune Function, and Intestinal Health in Broilers

**DOI:** 10.3390/ani14202926

**Published:** 2024-10-11

**Authors:** Yu-Yun Gao, Xiao-Ping Liu, Ying-Huan Zhou, Jia-Yi He, Bin Di, Xian-Yue Zheng, Ping-Ting Guo, Jing Zhang, Chang-Kang Wang, Ling Jin

**Affiliations:** 1China National Engineering Research Center of JUNCAO Technology, Fujian Agriculture and Forestry University, Fuzhou 350002, China; gaoyuyun2000@163.com; 2College of Animal Science, Fujian Agriculture and Forestry University, Fuzhou 350002, China; xiaopingliu2412@163.com (X.-P.L.); yh980125@163.com (Y.-H.Z.); hjy2010613@126.com (J.-Y.H.); 18362272307@163.com (B.D.); zhengxianyue2002@163.com (X.-Y.Z.); pingtingguo@fafu.edu.cn (P.-T.G.); jingzhang@fafu.edu.cn (J.Z.); wangchangkangcn@163.com (C.-K.W.)

**Keywords:** *Ganoderma lucidum* polysaccharide, feed additives, yellow-feather broilers, immune effect, intestinal barrier

## Abstract

**Simple Summary:**

Juncao-substrate *Ganoderma lucidum* Residue (HWE-JGLR) is a dark brown powder. It is obtained by cultivating *G. lucidum* using mycorrhizae as culture material and extracting it from the remaining culture substrate after *G. lucidum* harvesting. The application of HWE-JGLR in animal feeds not only reduces the waste of resources but also promotes the reuse of agricultural resources and sustainable development. In our study, HWE-JGLR not only optimised the growth performance of broilers but also enhanced their immune responses. Specifically, dietary supplementation with HWE-JGLR was associated with increased production and gene expression of the immunomodulatory cytokine interleukin-4 (IL-4) and IL-10. Meanwhile, levels and gene expression of the pro-inflammatory cytokine IL-1β were significantly reduced. In addition, HWE-JGLR enhanced the intestinal barrier aspect in poultry. The addition of HWE-JGLR upregulated the gene expression of the broiler jejunal mucosal tight junction proteins *Claudin*-*1* and *ZO*-*1*.

**Abstract:**

The purpose of this experiment was to investigate the effects of Hot Water Extract of Juncao-substrate *Ganoderma lucidum* Residue (HWE-JGLR) on the immune function and intestinal health of yellow-feather broilers. In an animal feeding experiment, 288 male yellow-feather broilers (1 day old) were randomly allocated to four treatment groups with six replicates of 12 birds each. The control (CON) group was fed a basal diet. HJ-1, HJ-2, and HJ-3 were fed a basal diet supplemented with 0.25%, 0.50%, and 1.00% HWE-JGLR, respectively. The feeding trial lasted for 63 d. The results showed increased ADFI (*p* = 0.033) and ADG (*p* = 0.045) of broilers in HJ-3, compared with the CON group. Moreover, higher contents of serum IL-4 and IL-10 and gene expression of IL-4 and IL-10 in jejunum mucosa and lower contents of serum IL-1β and gene expression of IL-1β in jejunum mucosa in HJ-3 were observed (*p* < 0.05). Additionally, the jejunal mucosal gene expression of *Claudin-1* and *ZO*-*1* in HJ-2 and HJ-3 was higher than that in the CON group (*p* < 0.05). As for the microbial community, compared with the CON group, the ACE index, Shannon index, and Shannoneven index of cecal microorganisms in HJ-2 and HJ-3 were elevated (*p* < 0.05). PCoA analysis showed that the cecal microbial structure of broilers in HJ-2 and HJ-3 was different from the CON group (*p* < 0.05). In contrast with the CON group, the broilers in HJ-2 and HJ-3 possessed more abundant Desulfobacterota at the phylum level and *unclassified Lachnospiraceae*, *norank Clostridia vadinBB60 group* and *Blautia* spp. at the genus level, while *Turicibacter* spp. and *Romboutsia* spp. were less (*p* < 0.05). In conclusion, dietary supplementation with HWE-JGLR can improve growth performance, enhance body immunity and intestinal development, and maintain the cecum microflora balance of yellow-feather broilers.

## 1. Introduction

*Ganoderma lucidum* (*G. lucidum*), a traditional Chinese medicine with a long history, derives its medicinal value from a wealth of bioactive components, primarily including polysaccharides, triterpenoids, sterols, lipids, and a variety of amino acids [1,2]. It is noteworthy that after the fruiting bodies of *Ganoderma* are harvested, the mycelium remaining in the cultivation substrate still harbors valuable resources [3]. In particular, the secondary metabolites produced by this mycelium, namely *Ganoderma lucidum* polysaccharides (GLP) [4]. GLP consists of the polymerization of a number of monosaccharides, such as glucose, mannose, arabinose, xylose, fucose, glucuronic acid, and galacturonic acid [5]. Hot Water Extract of Juncao-substrate *Ganoderma lucidum* Residue (HWE-JGLR) is extracted through a special process. The researchers chose Juncao as the culture substrate to cultivate *G. lucidum* until the *G. lucidum* fruiting entity matures and is harvested. During this process, while *G. lucidum* draws nutrients from the Juncao, its mycelium penetrates deep into the substrate and interacts with the raw material of the Juncao, potentially producing and accumulating unique bioactive. HWE-JGLR is a dark brown powdery substance that was finally obtained by Liu et al. [6] after using the hot water extraction method to process the leftover substrate and maximize the retention of the active ingredients therein. Previous studies showed that HWE-JGLR may play a positive role in enhancing the production performance and strengthening the immune function of dairy cows through its rich bioactive ingredients [6,7]. Our previous results showed that supplementation of HWE-JGLR in the diet could regulate lipid metabolism, enhance antioxidant capacity and improve intestinal health in 21-day-old yellow-feather broilers [8].

GLP, as an active ingredient in HWE-JGLR, has shown remarkable efficacy in regulating the immune system and improving gut microbiota [9,10]. GLP can effectively regulate immune cells and release a variety of cytokines to regulate the body’s immune response [11,12]. For example, GLP can elevate the IgG level in mouse serum and enhance the proliferation of splenocytes while elevating the levels of key immunomodulatory factors such as interleukin(IL)-4, IL-6, and interferon-γ (IFN-γ) in mice, enhancing the body’s ability of humoral and cellular immunity [13]. In addition, a study by Song et al. [14] found that GLP promoted the expression of the pro-inflammatory factor tumour necrosis factor-α (TNF-α) and IL-1β in macrophages, further enhancing the immune response. Moreover, GLP can also improve the body’s intestinal environment and regulate the intestinal microflora. GLP not only promotes beneficial bacteria such as *Lactobacillus*, *Propionibacterium*, and *Bifidobacterium* but also inhibits the growth of harmful bacteria such as *Corynebacterium*, *Streptomyces*, and *Proteus*, thereby contributing to the maintenance of intestinal microecological balance [4,15]. Chen et al. [15] further revealed that GLP can adjust the structure of the rat’s gut microbiota, GLP can reduce the relative abundance of Bacteroidetes and Proteus and increase the relative abundance of beneficial bacteria such as thick-walled bacilli, Lactobacillus and Ruminococcus in rats. Khan et al. [16] also revealed that GLP can modulate SFCA levels, which are crucial for maintaining the integrity of the gut barrier and nutrient absorption.

The gut health of poultry is a key factor in enhancing their production performance [17]. However, due to the overuse of antibiotics in recent years, poultry gut health has been negatively impacted [18]. As consumer health concerns continue to grow and awareness of antibiotic resistance issues increases, the need to find products that can replace antibiotics to maintain broiler health has become particularly urgent [19]. Polysaccharides obtained from *G. lucidum* have a wide range of pharmacological effects, such as antioxidant, immunomodulatory, anti-inflammatory, and antimicrobial properties [20]. However, the high cost of *G. lucidum* itself has limited its popularity in practical production applications. In contrast, HWE-JGLR extracted from the remaining medium was inexpensive and enriched in GLP. Therefore, we hypothesized that the addition of HWE-JGLR to the feed of yellow-feathered broilers may produce a similar effect as the addition of GLP. This experiment was conducted to investigate the feasibility of applying HWE-JGLR in practical production by determining the effects of HWE-JGLR on growth performance, immune function and intestinal health of yellow-feather broilers. The development of HWE-JGLR feed resources is of great significance for promoting the recycling of agricultural resources and alleviating ecological pressure.

## 2. Materials and Methods

### 2.1. Experimental Design, Animals, and Diets

A total of 288 1-day-old male yellow-feather broilers (average weight 34.10 ± 0.08 g, purchased from Fujian Wenshi Poultry Co., Ltd., Fuzhou, China) were randomly divided into four treatments with six replicates each (12 birds per replicate). The feedstuff used in this experiment was provided by Jinhualong Feed Factory (Fuzhou, China). HWE-JGLR includes GLP (15.79%), crude protein (23.58%), and crude fat (0.20%), among other components. GLP content was determined to be 15.79% following the procedure described by Nataraj et al. [21]. As our previous studies have said, the main active ingredient of HWE-JGLR is GLP, and other components are mainly carriers, including stone powder, silica, and lignin, which do not affect the growth performance of animals [8]. It is provided by the National Engineering Research Center of JUNCAO Technology (Fuzhou, China). The CON group was fed a corn-soybean basal diet, and the HJ-1, HJ-2, and HJ-3 were fed a basal diet supplemented with 0.25%, 0.50%, and 1.00% of HWE-JGLR for 63 days, respectively. The experiment was divided into three stages of ration formulation: 1 to 21 d, 22 to 42 d, and 43 to 63 d. The diets were formulated according to the Chinese Feeding Standard of Chicken (NY/T33-2004) [22]. The composition and nutrition level of the basal diet is shown in Table 1, Table 2 and Table 3. Six nutrients of corn, soybean meal and expanded soybeans in the formula were supplemented in Appendix A.

In the process of raising, broilers were housed in individual cages with 1562.5 cm² per bird. The feed ingredients less than 0.1% in the diet formula were first stirred evenly with the premix, and other feed ingredients greater than 0.1% were poured into the blender and stirred evenly, and then all the feeds were mixed in the blender. Each 50 kg feed needs to be mixed for a 30 min mixer from Changsha Ruijia Xuefeng Machinery Equipment Co., Ltd. (Changsha, Hunan, China). To prevent mildew in the feed, we will configure the feed one week before the start of a phase. The cage is equipped with LED lamps and implements 23 h of light per day and 1 h of dark cycle, and the light intensity is maintained at 50 lx. The temperature setting started at the initial 35 °C, decreased by 2–3 °C per week until it dropped to 22 °C, and maintained this temperature condition until the end of the experiment. At the same time, ensure that the henhouse has a good ventilation system. On day 5 of broiler growth, the chickens were immunized with infectious avian influenza, followed by Newcastle disease vaccination on the 10th and 20th days, respectively. All birds had ad libitum access to feed and water. All the experimental procedures applied in this study were reviewed and approved by the Committee of Animal Experiments of Fujian Agriculture and Forestry University (Fuzhou, Fujian, China, approval ID PZCASFAFU22009). The ethical approval date is 15 July 2021.

### 2.2. Growth Performance Measurement

A group of 288 1-day-old broilers was randomly selected, and the broilers were weighed and grouped afterward. Then, the average weight was calculated as the initial body weight (0.01 g precision electronic scale). After that, each replicate was weighed on day 63 and recorded as final body weight (0.1 g precision electronic scale). Average daily gain (ADG) was calculated as (total weight on day 63 − initial weight)/63. Feed intake was calculated by recording feed weight and residual feed weight for 63 d (0.01 g precision electronic scale), and the value of (total feed weight − total residual feed weight)/63 was used as the average daily feed intake (ADFI), and the feed gain ratio (F/G) was calculated based on ADG and ADFI.

### 2.3. Sampling Procedure

On day 63 of the experiment, two healthy yellow-feather broilers close to the average weight were randomly selected from each replicate, and the body weight before slaughter was recorded. After 12 h of fasting, blood samples were obtained from the broilers by puncturing their wing vein. The blood samples collected were centrifuged at 4 °C at 1000× *g* for 15 min. The supernatant was then collected, and the serum was transferred to −80 °C for further analysis. Approximately 3 cm of tissue from the duodenum, jejunum, and ileum was collected. The contents of the intestine were washed out using a normal saline solution, and then the duodenum, jejunum, and ileum tissue was preserved by fixing it in a 4% formaldehyde solution. Then, three sections of approximately 3 cm of mid-jejunal tissue were taken, and the mucosa was scraped off and frozen at −80 °C to detect digestive enzyme activity and gene expression of tight junction protein. Cecal digesta were collected and stored at −80 °C for 16S rRNA gene sequencing. The spleen, thymus, and bursa of Fabricius were taken and accurately weighed and recorded to calculate the immune organ indexes.

### 2.4. Determination of Digestive Enzyme Activities

The jejunum mucosa and digesta samples were thawed and homogenized in 0.9% NaCl solution, which was nine times larger in volume than the determining samples. Then, the homogenates were centrifuged at 1000× *g* for 10 min at 4 °C to obtain the supernatants. Total protein content in the intestinal mucosa was assessed by using a BCA protein quantitative kit purchased from New Cell § Molecular Biotech Co. Ltd. (Suzhou, China). The activities of lipase (Catalog number: A054-1-1), α-amylase (Catalog number: C016-1-1), chymotrypsin (Catalog number: A080-3-1) and trypsin (Catalog number: A080-2-2) were measured by commercial assay kits (Nanjing Jiancheng Bio-Engineering Institute, Nanjing, China) according to the instructions of the manufacturer [23]. The absorbance was detected by spectrophotometry (iMark, Bio-Rad, Hercules, CA, USA).

### 2.5. Intestinal Morphology

The fixed duodenum, jejunum, and ileum tissue were subjected to hematoxylin and eosin (H&E) staining. The tissue section was examined using a light microscope (Hitachi, Tokyo, Japan). Image-Pro Plus 6.0 software (Media Cybernetics Inc., Bethesda, MD, USA) was used to measure the villus height (VH), crypt depth (CD), and villus height/crypt depth (V/C).

### 2.6. Gene Expression Analysis by qPCR

Total RNA was extracted from the jejunal mucosa by using the TransZol UP Plus Total RNA Extraction Kit (Catalog number: ER501-01-V2) from Beijing TransGen Biotech Co., Ltd. (Beijing, China) and a Thermo Scientific NanoDrop 2000 Spectrophotometer (Thermo Fisher Scientific, Wilmington, NC, USA) was used to determine the concentration and quality of RNA. Total RNA was reverse-transcribed by using the Synthesis SuperMix for qPCR from Beijing TransGen Biotech Co., Ltd. (Beijing, China). The cDNA was used for quantitative real-time PCR on Applied Biosystems. The relative mRNA expression levels of each target gene were analyzed according to the method described by Livak and Schmittgen (2001) and shown as 2^−ΔΔCT^. The primer sequences are shown in Table 4.

### 2.7. Bacterial DNA Extraction and 16S rRNA Gene Sequencing

Cecal contents were collected sterilely and preserved in liquid nitrogen before storage at −80 °C. Total fecal DNA was extracted using the Stool DNA Kit (D4015-01, Omega Bio-tek, Norcross, GA, USA), and the purity and concentration of extracted DNA were assessed by measuring the OD 260/280 and OD 260/230 ratios. DNA was stored at −80 °C for subsequent analysis. For the analysis of microbial diversity and composition of the cecum, we selected the highly variable V3-V4 region of the bacterial 16S rRNA gene as the target region, which was amplified by PCR using specific primers 338F (5′-ACTCCTACGGGAGGCAGCAG-3′) and 806R (5′-GGACTACHVGGGTWTCTAAT-3′) by an ABI GeneAmp 9700 PCR thermocycler (Waltham, MA, USA). Subsequently, the amplification products were purified and sent to the Illumina MiSeq PE300 platform (Illumina, Inc, San Diego, CA, USA) for high-throughput sequencing, a process that was performed by Majorbio Bio-Pharm Technology Co., Ltd. (Shanghai, China) according to established methods to obtain high-quality sequence data. The Venn diagram illustrates unique and shared Amplicon Sequence Variants (ASVs) across groups. Alpha diversity was assessed using accumulated cyclone energy (ACE), Shannon, and Shannoneven diversity indices with the Mothur software (V1.45.3) [24]. Principal coordinate analysis (PCoA) based on Bray–Curtis distances to evaluate sample and group similarities, while intergroup differences were determined by the Analysis of Similarities (ANOSIM) test.

### 2.8. Data Analysis and Statistics

Statistical analyses were conducted using SPSS, version 22.0 (SPSS, Inc., Chicago, IL, USA). The Shapiro–Wilk test was used to check the normality of all data, and Levene’s test was used to check the homogeneity of variance. Data were evaluated using one-way analysis of variance (ANOVA) with Tukey’s multiple range test, and a *p*-value < 0.05 was considered statistically significant.

## 3. Results

### 3.1. Growth Performance

The effects of HWE-JGLR on growth performance are presented in Table 5. Compared with the CON group, the ADFI (*p* = 0.033) and ADG (*p* = 0.045) of broilers supplemented with 0.50% and 1.00% HWE-JGLR increased, and there was a highly significant effect of FBW (*p* < 0.001) in broilers with 0.25%, 0.50% and 1.00% HWE-JGLR. But there was no difference in IBW, Mortality and F/G (*p* > 0.05).

### 3.2. Immune Indicators

Dietary supplementation with HWE-JGLR had no effect on the immune organ index of 63-day-old broilers, as shown in Table 6 (*p* > 0.05).

Compared with the CON group, the serum contents of IL-4 and IL-10 in HJ-3 increased, and the serum IL-1β content of HJ-3 decreased (*p* < 0.05). There were no differences in serum IFN-γ, IL-6, IgA, IgG, and IgM of broilers among groups (Figure 1A,B).

### 3.3. Gene Expression of Cytokines in the Jejunum Mucosa

As shown in Figure 2A, compared with the CON group, the gene expression of IL-1β in the jejunal mucosa of broilers in HJ-1, HJ-2, and HJ-3 decreased. The gene expression of the IL-10 in the jejunal mucosa of broilers in HJ-1, HJ-2, and HJ-3 strengthened, and the gene expression of IL-4 in the jejunal mucosa in HJ-2 and HJ-3 increased (*p* < 0.05). There were no differences in small intestine mucosal IFN-γ and IL-6 gene expression among the groups (*p* > 0.05).

### 3.4. Gene Expression of Tight Junction Protein in the Jejunum Mucosa

As shown in Figure 2B, compared with the CON group, the gene expression of *Claudin-1* in the jejunal mucosa of broilers enhanced in HJ-3, and the gene expression level of *ZO-1* in the jejunal mucosa increased in HJ-2 and HJ-3 (*p* < 0.05). The gene expression of *Occludin* in the jejunum mucosa was not different among groups (*p* > 0.05).

### 3.5. Digestive Enzyme Activity

As shown in Table 7, there were no differences in jejunal digestive enzyme activities among groups (*p* > 0.05).

### 3.6. Jejunum Histomorphology

As shown in Table 8, no changes in jejunal VH, CD, and V/C of broilers were observed among groups (*p* > 0.05).

### 3.7. Cecal Microbiota

After denoise and quality control, 499,848 16S rRNA gene sequences were obtained from 24 cecal bacterial DNA samples of broilers, and a total of 5851 ASVs were obtained based on the DADA2 modeling algorithm. Rarefaction curve analysis showed that the sequencing data were sufficient to reflect the microbial diversity (Figure 3A). On day 63, there were 882 shared ASVs among groups, whereas 48, 79, 82, and 94 ASVs were identified as only present in the CON group, HJ-1, HJ-2, and HJ-3, respectively (Figure 3B). As shown in Table 9, the ACE index, Shannon index and Shannoneven index of cecum microflora were increased in HJ-2 and HJ-3 compared with the CON group (*p* < 0.05). As revealed in Figure 3C, PCoA based on the Bray–Curtis distance showed that the distance of samples between groups is far (*p* = 0.001), indicating that the microbial structure between groups was distinguishing. As presented in Table 10 and Figure 3D, at the phylum level, the cecal microbiota community of broilers was dominated by Firmicutes, Bacteroidota, Desulfobacterota, Actinobacteriota, and Proteobacteria, and their proportions accounted for more than 99%. The relative abundance of Desulfobacterota in HJ-3 was increased compared with the CON group (*p* < 0.05). At the genus level, the cecal microbiota community was dominated by *Romboutsia* spp., *unclassified Lachnospiraceae*, *Turicibacter* spp., *norank Clostridia vadinBB60 group*, *norank Clostridia UCG-014*, *Faecalibacterium* spp., *Blautia* spp., *Ruminococcus torques group*, and *Lactobacillus* spp. The relative abundances of *Romboutsia* spp. in HJ-2 and HJ-3 decreased compared with the CON group (*p* < 0.05). The relative abundances of *unclassified Lachnospiraceae* and *Blautia* spp. in HJ-2 and HJ-3 enhanced (*p* < 0.05). The relative abundance of *Turicibacter* spp. in HJ-2 decreased (*p* < 0.05). The relative abundance of *norank Clostridia vadinBB60 group* among HJ-1, HJ-2, and HJ-3 elevated (*p* < 0.05). The relative abundances of *Ruminococcus torques group* in HJ-2 upregulated (*p* < 0.05) (Table 10 and Figure 3E).

## 4. Discussion

In recent years, several studies have shown that GLP has a beneficial effect on gut health and growth performance [25,26]. Liu et al. [27] further demonstrated that sporoderm-broken spores of *G. lucidum* could increase the ADG of poultry and could effectively reduce the F/G of broilers. Moreover, in the field of aquaculture, Mohan et al. [28] further proved that GLP has a positive effect on the production performance of shrimp, which increased ADG, AFDI, and survival rate of shrimp, and disease resistance was also upregulated. Following the results of previous studies, our experiment likewise found a positive effect of the HWE-JGLR on broiler growth performance, further corroborating the above findings. Our experiments showed that the addition of 0.50% and 1.00% of HWE-JGLR to the diet increased ADFI and ADG. To further analyze the mechanism of HWE-JGLR on the growth performance of broilers, we focused on the two dimensions of immune function and intestinal health of broilers.

Cytokines are small molecule proteins with various biological activities secreted by immune cells, which are important mediators in the immune response [29]. Normally, there is a dynamic balance of pro-inflammatory and anti-inflammatory factors in the body, ensuring proper regulation of the immune response [30]. For example, IL-1β acts as a pro-inflammatory factor and is involved in regulating the proliferation, differentiation, and apoptosis of cells involved [31]. As a key immunomodulatory molecule, IL-4 plays a central regulatory role in humoral and adaptive immunity by promoting the proliferation of activated B cells and T cells. On the other hand, IL-10 plays a key role in anti-inflammatory effects, effectively balancing the immune response by down-regulating the expression levels of pro-inflammatory cytokines such as TNF-α and IL-1β [32]. Our results showed that dietary additions of 0.25%, 0.50%, and 1.00% HWE-JGLR significantly altered immune indices in broilers. Specifically, it reduced IL-1β gene expression and enhanced IL-10 gene expression in the jejunal mucosa of broilers. The addition of 0.5% and 1% HWE-JGLR increased the levels of IL-4 content in the serum and the levels of IL-4 gene expression jejunal mucosa of broilers. This is consistent with the results of previous studies that added GLP that 400 mg/kg GLP reduced serum levels of IL-1β and IL-6 in diabetic rats [15], and 800 mg/kg GLP reduced serum IL-6 and TNF-α levels and increased serum IL-2, IL-4, and IL-10 levels in rats with gastric cancer [33]. On this basis, we speculate that the immunomodulatory mechanism of HWE-JGLR may involve direct activation of immune cells and act through specific receptor pathways. It has been revealed that GLP can activate the NF-κB and p38 MAPK signalling pathways by binding to TLR4 on the surface of macrophages, triggering a series of immune responses, including the release of cytokines such as TNF-α [34]. In addition, the interaction of GLP with pattern recognition receptors (PRRs) promotes the synthesis and release of immune-related molecules such as cytokines, chemokines, and nitric oxide (NO), further activating and directing immune cell functions [11]. Consequently, we infer that HWE-JGLR may interact with receptors on the membrane of immune cells, activating signaling pathways such as NF-κB and p38 MAPK, which in turn regulate the expression of cytokines. This mechanism effectively adjusts the immune balance in broilers, leading to improved growth performance.

The integrity of the intestinal barrier and the stability of the intestinal flora are key factors affecting the intestinal health of broilers. Healthy intestinal status can directly enhance broiler performance by improving nutrient absorption efficiency, maintaining intestinal microbial balance, and enhancing immune response [35]. As a natural bioactive substance, GLP has been widely studied and proven to have a positive effect on regulating intestinal microbial communities, maintaining intestinal homeostasis, and promoting intestinal health [16,36,37]. Sang et al. [36] found that 300 mg/kg GLP elevated gene expression of *Occludin*, *Claudin-1*, and *ZO-1* in mice on a high-fat diet. We generally consider that upregulation of *Claudin-1* and *ZO-1* gene expression effectively maintains cell polarity, intercellular adhesion fixation, and ion transport in cellular bypass in epithelial and endothelial cells [38]. Consequently, we infer that the HWE-JGLR potentially fosters a positive influence on the intestinal barrier function and the stability of the gut microbiota in broilers. As we expected, in terms of intestinal barrier function, the addition of 0.5% and 1% HWE-JGLR increased gene expression of *ZO-1* in broiler jejunal mucosa, and 1.00% of HWE-JGLR elevated jejunal mucosa *Claudin-1* gene expression. In maintaining the homeostasis of gut microbiota, Ren et al. [39] found that GLP increased the diversity of gut microbial communities in high-diet mice and increased the abundance of beneficial bacteria such as *Mycobacterium* spp. Chen et al. [15] found that 400 mg/kg GLP reduced the abundance of harmful bacteria such as Aerococcus, Rumenococcus, Corynebacterium, and Aspergillus in type 2 diabetic rats. Additionally, Guo et al. [40] demonstrated that GLP ameliorated microbiota dysbiosis and increased the abundance of beneficial Bifidobacterium and Lactobacillus strains as well as the production of SCFAs by inhibiting TLR4/MyD88/NF-κB signaling, which attenuated endotoxaemia. Therefore, we infer that HWE-JGLR may play a role in enhancing the abundance and diversity of microbial communities in the broiler cecum, increasing the abundance of beneficial intestinal bacteria and inhibiting the colonisation of harmful bacteria. As we expected, our results showed that dietary supplemented with 0.50% and 1.00% HWE-JGLR increased the microbial α-diversity and affected the microbial composition of broilers. *Blautia* spp. is an anaerobic bacterium that produces bacteriocins to prevent pathogen colonization, efficiently promotes metabolism and regulates host health by upregulating T cell and SCFA production [41]. Our results showed that dietary supplemented with 0.50% and 1.00% HWE-JGLR increased the relative abundance of *Blautia* spp. in the cecum of broilers. In addition, we found that the relative abundance of *Turicibacter* spp. in the cecum of broilers was decreased by the supplement of HWE-JGLR. Liang et al. [42] showed that *Turicibacter* spp. was positively correlated with pro-inflammatory cytokines. Thus, the decrease in the relative abundance of *Turicibacter* spp. may be due to a decrease in the expression of the pro-inflammatory factor IL-1β due to the addition of HWE-JGLR. In conclusion, HWE-JGLR can indirectly regulate the pro-inflammatory state in the gut to maintain the homeostasis of the intestinal flora by increasing the abundance of beneficial bacteria and decreasing the abundance of potentially harmful bacteria.

## 5. Conclusions

In conclusion, dietary supplementation of HWE-JGLR can enhance the production of immuno-modulatory cytokines IL-4 and IL-10 and upregulate their gene expression. At the same time, HWE-JGLR could significantly reduce the level and gene expression of pro-inflammatory cytokine IL-1β. In addition, the addition of HWE-JGLR upregulated the gene expression of tight junction proteins *Claudin-1* and *ZO-1* in the jejunum mucosa of broilers and strengthened the intestinal barrier of broilers. It can effectively regulate the intestinal flora and increase the number of beneficial bacteria. Therefore, it is recommended to supplement 1.00% HWE-JGLR in broiler diets. All results indicate that HWE-JGLR may be a good feed additive that can replace GLP extracted from *G. lucidum* fruiting bodies and is a new functional additive in the production of yellow-feather broilers.

## Figures and Tables

**Figure 1 animals-14-02926-f001:**
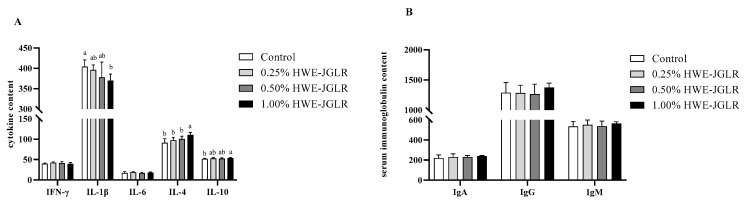
Effects of HWE-JGLR on serum cytokines (**A**) and immunoglobulins (**B**) contents of broilers. Values are presented as a mean ± SD. Letters in the case of each cytokine describe significant differences between groups at *p* < 0.05.

**Figure 2 animals-14-02926-f002:**
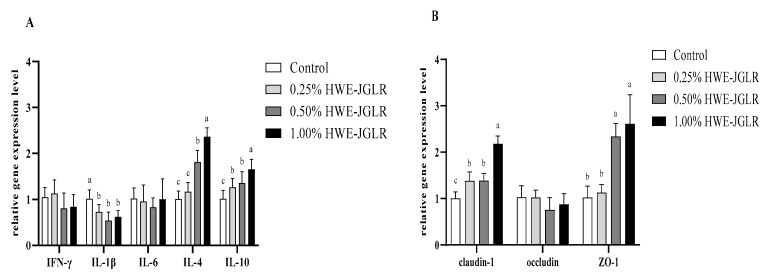
Effects of HWE-JGLR on gene expression of jejunal mucosal cytokine (**A**) and jejunal tight-junction protein (**B**) in broilers. Values are presented as a mean ± SD. Letters in the case of each cytokine or jejunal tight-junction protein describe significant differences between groups at *p* < 0.05.

**Figure 3 animals-14-02926-f003:**
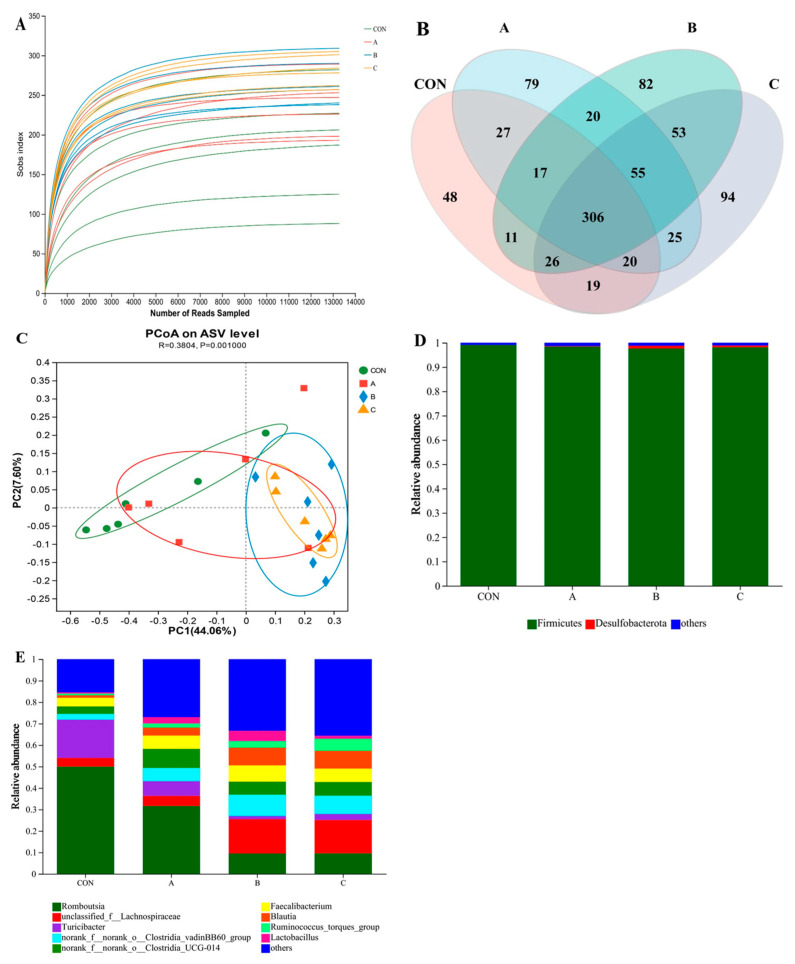
Rarefaction curves of sequencing read (**A**) and Venn diagram (**B**) of ASVs in broilers. PCoA analysis based on the Bray-Curtis distance (**C**). Relative abundance at phylum level (**D**) and genus level (**E**) in cecum microbiota. In (**B**–**D**) A represents HJ-1, B represents HJ-2, and C represents HJ-3.

**Table 1 animals-14-02926-t001:** Basal diet composition and nutrient levels from 1 to 21 d (air-dry basis).

Items	Groups			
	CON	HJ-1	HJ-2	HJ-3
Ingredients, %				
Corn	58.03	57.70	57.38	56.71
Soybean meal, 46% CP	32.64	32.57	32.48	32.35
Expanded soybean	4.00	4.00	4.00	4.00
Soybean oil	1.16	1.30	1.45	1.75
Limestone powder	1.12	1.12	1.12	1.12
Dicalcium phosphate	1.57	1.57	1.57	1.58
DL-Methionine, 98%	0.18	0.18	0.19	0.19
HWE-JGLR	0.00	0.25	0.50	1.00
Premix ^1^	1.00	1.00	1.00	1.00
NaCl	0.30	0.30	0.30	0.30
Total	100.00	100.00	100.00	100.00
Nutrient levels ^2^				
Metabolizable energy, MJ/kg	12.12	12.12	12.12	12.12
Crude protein, %	21.00	21.00	21.00	21.00
Calcium, %	1.00	1.00	1.00	1.00
Non-phytate phosphorus, %	0.45	0.45	0.45	0.45
Lysine, %	1.10	1.10	1.10	1.09
Methionine + Cystine, %	0.85	0.85	0.85	0.85

^1^ Nutrient levels of premix in Phase 1 (per kg diet): vitamin A, 5000 IU; vitamin D_3_, 1000 IU; vitamin E, 10 IU; vitamin K_3_, 0.5 mg; vitamin B_1_, 1.3 mg; vitamin B_2_, 3.6 mg; vitamin B_6_, 2.5 mg; vitamin B_12_, 0.01 mg; nicotinic acid, 35 mg; pantothenic acid, 10 mg; biotin, 0.15 mg; folic acid, 0.55 mg; choline chloride, 1000 mg; Cu (sulfate), 8 mg; Fe (sulfate), 80 mg; Mn (sulfate), 80 mg; Zn (sulfate), 60 mg; I (iodide), 0.35 mg; Se (selenite), 0.15 mg. ^2^ Nutrient levels were calculated values.

**Table 2 animals-14-02926-t002:** Basal diet composition and nutrient levels from 22 to 42 d (air-dry basis).

Items	Groups			
	Con	HJ-1	HJ-2	HJ-3
Ingredients, %				
Corn	62.89	62.56	62.23	61.57
Soybean meal, 46% CP	26.45	26.38	26.31	26.17
Expanded soybean	5.00	5.00	5.00	5.00
Soybean oil	1.83	1.97	2.12	2.42
Limestone powder	1.04	1.04	1.04	1.04
Dicalcium phosphate	1.38	1.38	1.38	1.39
DL-Methionine, 98%	0.11	0.11	0.11	0.11
HWE-JGLR	0.00	0.25	0.50	1.00
Premix ^1^	1.00	1.00	1.00	1.00
NaCl	0.30	0.30	0.30	0.30
Total	100.00	100.00	100.00	100.00
Nutrient levels ^2^				
Metabolizable energy, MJ/kg	12.54	12.54	12.54	12.54
Crude protein, %	19.00	19.00	19.00	19.00
Calcium, %	0.90	0.90	0.90	0.90
Non-phytate phosphorus, %	0.40	0.40	0.40	0.40
Lysine, %	0.97	0.97	0.97	0.96
Methionine + Cystine, %	0.73	0.73	0.72	0.72

^1^ Nutrient levels of premix in Phase 2 (per kg diet): vitamin A, 5000 IU; vitamin D_3_, 1000 IU; vitamin E, 10 IU; vitamin K_3_, 0.5 mg; vitamin B_1_, 1.3 mg; vitamin B_2_, 3.6 mg; vitamin B_6_, 2.5 mg; vitamin B_12_, 0.01 mg; nicotinic acid, 30 mg; pantothenic acid, 10 mg; biotin, 0.15 mg; folic acid, 0.55 mg; choline chloride, 750 mg; Cu (sulfate), 8 mg; Fe (sulfate), 80 mg; Mn (sulfate), 80 mg; Zn (sulfate), 60 mg; I (iodide), 0.35 mg; Se (selenite), 0.15 mg. ^2^ Nutrient levels were calculated values.

**Table 3 animals-14-02926-t003:** Basal diet composition and nutrient levels from 43 to 63 d (air-dry basis).

Items	Groups			
	Con	HJ-1	HJ-2	HJ-3
Ingredients, %				
Corn	71.22	70.97	70.60	69.92
Soybean meal, 46% CP	18.57	18.47	18.42	17.45
Expanded soybean	4.00	4.00	4.00	5.00
Soybean oil	2.48	2.58	2.74	2.86
Limestone powder	1.00	1.00	1.00	1.00
Dicalcium phosphate	1.20	1.20	1.21	1.23
DL-Methionine, 98%	0.11	0.11	0.11	0.11
HWE-JGLR	0.00	0.25	0.50	1.00
Premix ^1^	0.30	0.30	0.30	0.30
NaCl	0.12	0.12	0.12	0.13
Total	100.00	100.00	100.00	100.00
Nutrient levels ^2^				
Metabolizable energy, MJ/kg	12.96	12.96	12.96	12.96
Crude protein, %	16.00	16.00	16.00	16.00
Calcium, %	0.80	0.80	0.81	0.81
Non-phytate phosphorus, %	0.35	0.34	0.35	0.35
Lysine, %	0.85	0.85	0.85	0.85
Methionine + Cystine, %	0.65	0.65	0.65	0.65

^1^ Nutrient levels of premix in Phase 3 (per kg diet): vitamin A, 5000 IU; vitamin D_3_, 1000 IU; vitamin E, 10 IU; vitamin K_3_, 0.5 mg; vitamin B_1_, 1.3 mg; vitamin B_2_, 3.0 mg; vitamin B_6_, 2.5 mg; vitamin B_12_, 0.01 mg; nicotinic acid, 25 mg; pantothenic acid, 10 mg; biotin, 0.15 mg; folic acid, 0.55 mg; choline chloride, 500 mg; Cu (sulfate), 8 mg; Fe (sulfate), 80 mg; Mn (sulfate), 80 mg; Zn (sulfate), 60 mg; I (iodide), 0.35 mg; Se (selenite), 0.15 mg. ^2^ Nutrient levels were calculated values.

**Table 4 animals-14-02926-t004:** Primer sequences and amplification parameters of reference and target genes.

Gene	Primer Sequence 5′→3′	GenBank No.	Product Length/bp
*β-Actin*	F: GAGAAATTGTGCGTGACATCA R: CCTGAACCTCTCATTGCCA	NM_205518	152
*IFN-γ*	F: CTTCCTGATGGCGTGAAGA R: GAGGATCCACCAGCTTCTGT	NM_205149.2	127
*IL-1β*	F: GGAGCAGGGACTTTGCTGAC R: AAGGACTGTGAGCGGGTGTA	NM_204524.2	130
*IL-10*	F: GCTGTCACCGCTTCTTCAC R: TCACTTCCTCCTCCTCATCAG	NM_001004414.2	164
*IL-6*	F: CCTCCTCGCCAATCTGAAGT R: GCACTGAAACTCCTGGTCTTT	NM_204628.1	138
*IL-4*	F: TCTTCCTCAACATGCGTCAG R: TGGTGGAAGAAGGTACGTAGG	NM_001007079.1	127
*Claudin-1*	F: ATGGTATGGCAACAGAGTG R: CAGGAGCAGCAGAGGAAT	NM_001013611	144
*Occludin*	F: GATGTCCAGCGGTTACTACTAC R: GAAGAAGCAGATGAGGCAGAG	XM_025144248	172
*ZO-1*	F: TGCTTCCAGTGCCAACAGA R: CTTGCCAACCGTAGACCATAC	XM_015278981	183

**Table 5 animals-14-02926-t005:** Effects of HWE-JGLR on growth performance of broilers ^1^.

Items	Groups				*p*-Value
	CON	HJ-1	HJ-2	HJ-3	
IBW, g	34.32 ± 0.17	34.31 ± 0.15	34.25 ± 0.06	34.34 ± 0.12	0.072
FBW, g	2297.50 ± 1.87 ^a^	2303.50 ± 1.87 ^b^	2309.50 ± 1.87 ^c^	2315.50 ± 1.87 ^d^	<0.001
ADFI, g	80.29 ± 2.61 ^b^	82.13 ± 2.11 ^ab^	81.51 ± 0.86 ^b^	84.71 ± 3.61 ^a^	0.033
ADG, g	34.37 ± 1.73 ^b^	35.79 ± 0.56 ^ab^	36.02 ± 0.96 ^a^	36.32 ± 1.16 ^a^	0.045
F/G	2.34 ± 0.06	2.31 ± 0.03	2.28 ± 0.03	2.33 ± 0.08	0.256
Mortality, %	5.56	1.39	2.78	2.99	0.549

^1^ Values are means of six replicates per treatment with 12 birds each (*n* = 6). IBW = Initial body weight; FBW = Final body weight; ADFI = average daily feed intake; ADG = average daily gain; F/G = ratio of feed to gain. ^a–d^ Means in the same row without the same superscript significantly differ at *p* < 0.05. The data are presented as mean ± SD.

**Table 6 animals-14-02926-t006:** Effects of HWE-JGLR on immune organ indexes of broilers (g/kg) ^1^.

Items	Groups				*p*-Value
	CON	HJ-1	HJ-2	HJ-3	
Spleen index	1.19 ± 0.33	1.10 ± 0.23	1.04 ± 0.25	1.07 ± 0.20	0.223
Thymus index	2.42 ± 1.49	1.56 ± 1.01	2.57 ± 1.10	2.33 ± 1.34	0.074
Bursa of Fabricius index	1.70 ± 0.40	1.54 ± 0.58	1.55 ± 0.63	1.56 ± 0.62	0.549

^1^ Values are means of six replicates per treatment with 12 birds each (*n* = 6). The data are presented as mean ± SD.

**Table 7 animals-14-02926-t007:** Effects of HWE-JGLR on jejunal digestive enzyme activities of broilers ^1^.

Items	Groups				*p*-Value
	CON	HJ-1	HJ-2	HJ-3	
Lipase, U/gprot	197.83 ± 34.77	190.93 ± 56.83	200.84 ± 25.77	206.10 ± 88.24	0.983
Trypsin, U/mgprot	14,923 ± 2567	15,091 ± 1914	17,215 ± 2477	14,945 ± 2987	0.365
Chymotrypsin, U/mgprot	29.37 ± 10.08	30.10 ± 5.48	32.87 ± 9.92	28.77 ± 6.79	0.840
α-amylase, U/gprot	0.25 ± 0.14	0.16 ± 0.05	0.23 ± 0.15	0.17 ± 0.12	0.611

^1^ Values are means of six replicates per treatment with 12 birds each (*n* = 6) The data are presented as mean ± SD.

**Table 8 animals-14-02926-t008:** Effects of HWE-JGLR on intestinal morphology of 63-day-old broilers ^1^.

Items		Groups				*p*-Value
		CON	HJ-1	HJ-2	HJ-3	
Duodenum	VH (μm)	990.50 ± 50.24	1049 ± 190.62	1022 ± 197.42	1125 ± 148.78	0.512
	CD (μm)	82.30 ± 18.54	81.66 ± 9.20	77.18 ± 4.23	83.02 ± 6.83	0.798
	V/C	12.77 ± 2.68	13.46 ± 4.19	13.46 ± 2.42	13.79 ± 1.73	0.940
Jejunum	VH (μm)	1026 ± 41.24	997.30 ± 89.18	1054 ± 54.52	1141 ± 137.59	0.058
	CD (μm)	93.52 ± 7.22	95.21 ± 9.30	89.75 ± 13.55	94.71 ± 9.53	0.787
	V/C	11.16 ± 1.21	10.57 ± 1.09	12.12 ± 2.25	12.18 ± 1.26	0.229
Ileum	VH (μm)	865.30 ± 111.47	966.30 ± 87.30	833 ± 167.37	875.10 ± 129.29	0.331
	CD (μm)	86.01 ± 9.34	79.60 ± 10.59	77.95 ± 8.19	73.09 ± 11.09	0.188
	V/C	10.18 ± 0.96	12.50 ± 2.43	11.04 ± 2.82	12.63 ± 2.93	0.264

^1^ Values are means of six replicates per treatment with 12 birds each (*n* = 6). The data are presented as mean ± SD.

**Table 9 animals-14-02926-t009:** Effects of water extract of HWE-JGLR on alpha diversity of cecum microorganism ^1^.

Items	Groups				*p*-Value
	CON	HJ-1	HJ-2	HJ-3	
ACE	188.80 ± 70.09 ^b^	235.80 ± 36.38 ^ab^	268.50 ± 27.34 ^a^	284.40 ± 19.81 ^a^	0.014
Shannon	2.73 ± 1.09 ^b^	3.71 ± 0.75 ^ab^	4.44 ± 0.31 ^a^	4.57 ± 0.07 ^a^	0.004
Shannoneven	0.52 ± 0.17 ^b^	0.68 ± 0.12 ^ab^	0.80 ± 0.04 ^a^	0.81 ± 0.01 ^a^	0.002

^1^ Values are means of six replicates per treatment with 12 birds each (*n* = 6). ^a,b^ Means in the same row without the same superscript significantly differ at *p* < 0.05. The data are presented as mean ± SD.

**Table 10 animals-14-02926-t010:** The relative richness of cecal dominant phyla and genera in 63-day-old broilers ^1^.

Items (%)	Groups				*p*-Value
	CON	HJ-1	HJ-2	HJ-3	
Phylum level					
Firmicutes	99.05 ± 1.35	98.39 ± 1.03	97.64 ± 1.00	98.13 ± 0.89	0.117
Bacteroidota	0.69 ± 0.99	0.99 ± 0.93	0.63 ± 0.44	0.58 ± 0.72	0.759
Desulfobacterota	0.05 ± 0.09 ^c^	0.20 ± 0.33 ^b^	1.10 ± 0.76 ^a^	0.71 ± 0.43 ^a^	0.002
Actinobacteriota	0.06 ± 0.10	0.25 ± 0.24	0.29 ± 0.18	0.45 ± 0.43	0.149
Proteobateria	0.13 ± 0.22	0.11 ± 0.12	0.26 ± 0.47	0.06 ± 0.08	0.816
Genus level					
*Romboutsia*	49.96 ± 19.41 ^a^	31.65 ± 22.58 ^ab^	9.51 ± 5.96 ^b^	9.56 ± 5.22 ^b^	0.006
*unclassified Lachnospiraceae*	4.04 ± 4.73 ^b^	4.69 ± 2.85 ^b^	15.96 ± 5.10 ^a^	15.54 ± 6.06 ^a^	0.001
*Turicibacter*	17.82 ± 17.55 ^a^	6.85 ± 6.29 ^ab^	1.46 ± 1.40 ^b^	2.80 ± 2.35 ^ab^	0.030
*norank Clostridia vadinBB60 group*	2.70 ± 2.54 ^b^	6.14 ± 4.81 ^a^	9.94 ± 2.18 ^a^	8.52 ± 4.91 ^a^	0.026
*norank Clostridia UCG-014*	3.49 ± 2.85	8.94 ± 2.69	6.11 ± 2.04	6.40 ± 1.77	0.293
*Faecalibacterium*	3.99 ± 3.50	6.15 ± 2.70	7.54 ± 4.95	6.22 ± 3.42	0.449
*Blautia*	1.18 ± 0.92 ^b^	3.74 ± 3.45 ^ab^	8.26 ± 4.73 ^a^	8.25 ± 3.89 ^a^	0.002
*Ruminococcus torques group*	0.79 ± 0.76 ^b^	1.89 ± 1.51 ^b^	3.06 ± 1.49 ^ab^	5.65 ± 3.23 ^a^	0.002
*Lactobacillus*	0.54 ± 0.76	2.92 ± 3.77	4.78 ± 9.63	1.37 ± 1.06	0.460

^1^ Values are means of six replicates per treatment with 12 birds each (*n* = 6). ^a–c^ Means in the same row without the same superscript significantly differ at *p* < 0.05. The data are presented as mean ± SD.

## Data Availability

Most of the data generated or analyzed in this study are presented in this published article or its Appendix A. Additional data not included here are accessible upon reasonable request to the corresponding author.

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
