# Peer review of "The Addition of Hot Water Extract of Juncao-Substrate Ganoderma lucidum Residue to Diets Enhances Growth Performance, Immune Function, and Intestinal Health in Broilers"

_animals, 2024, doi:10.3390/ani14202926_

Round 1
Reviewer 1 Report
Comments and Suggestions for Authors
The authors noted the importance and expediency of studying new phytobiotic feed additives in poultry diets in order to provide animals with immunosuppressive additives, which also have a positive effect on productivity and biochemical parameters. In my opinion, this is an actual and valuable addition for agriculture. Because it will help to replace more efficient and cheaper feeds with alternatives from a more profitable and economical side. In general, the work is relevant and of great interest for further development.
However, during the review process, some inaccuracies and recommendations to the authors were noted, to which we would like to receive a more detailed answer:
1. In the literature review, the authors indicate that the extract is obtained in a special way (line 54-55), but there is no reference to the source.
2. The literature review in this article meets the requirements. It lists modern literature and has all the necessary references.
3. The methodology meets the requirements. There is a protocol of the ethics commission.
4. The methodology contains
5. Line 140. It is not entirely clear what the authors meant. What does "randomly weighed chickens" mean? It is not clear how many animals were used in the calculation of meat productivity.
6. Line 143 check the formula. This is the first time I've seen an absolute daily increase measured as a percentage.
7. Line 158 you probably frozen the serum samples at a temperature of - 80 C.
8. What is the reason that you took a blood test from only 2 animals?
9. Also, in my humble opinion, the authors should have given the contents of the BAS. It was clear from the literature review that this extract is useful for both humans and animals. However, the authors did not conduct a quantitative or qualitative analysis of the characteristics of the product under study.
10. At the same time, the authors point out that this extract is rich in amino acids, but this does not seem to affect the composition of the diet in any way.
In general, concluding, it can be concluded that the publication of the manuscript deserves to be published after correcting the comments.
Author Response
请参阅附件

Reviewer 2 Report
Comments and Suggestions for Authors
In the manuscript, the authors studied The addition of Hot Water Extract of Juncao-substrate Ganoderma Lucidum Residue to diets enhances Growth Performance, Immune Function, and Intestinal Health in Broilers.
However, the following comments can be made.
1. Methodology. Why were only males used in the studies? What is the reason for this?
2. Methodology. There is no description of the procedure for introducing the test object into broiler diets. Describe the procedure.
3. Conclusion. Expand and describe in more detail!
Reviewer 3 Report
Comments and Suggestions for Authors
The purpose of this experiment was to investigate the effects of Hot Water Extract of Juncao-substrate Ganoderma Lucidum Residue (HWE-JGLR) on the immune function and intestinal health of yellow-feather broilers. The results obtained are important for poultry practice.
General comments:
In my opinion, the article should be supplemented with the following information:
In the Introduction chapter, there is no research hypothesis and no information about the innovative elements of this study.
In the Materials and methods chapter there is no information about:
a. environmental parameters: temperature, relative humidity, light intensity and color, type of light (incandescent, fluorescent?)
b . No information about the type of scale for recording BW, FI and measurement accuracy
c. No information on BW and % moratality in Table 5
Others
For significance, use a low letter "p" in italic instead of the capital "P" in the main article
The headings in the Tables: in the Materials and Methods and Results chapter must be in bold (see other articles in Animals)
The References section must be done according to the instructions for the authors
Please use a "dot" after each abbreviation, for example "Int. J. Biol.. Macromol. " instead of Int J Biol Micromol
Year in bold, full page range (855-862 instead of 855-62),
For page ranges use long dash (-) from the symbol function, instead of short (-) from the keyboard
Titles of works without quotation marks
Specific comments
L5-6 No affiliation designation for authors 3 to 9, for designations 1 and 2 use superscript
L11 no phone number
L29 ADFI only for HJ-3
L39 Turicibacter also less
L65 [3,4] no space after 3
L81 Chen et al. [10]
L81 [9,10]
L84 delete [10]
L84 Khan et al. [11]
L 86 delete [11]
L102 space before "g"
L114 (2004) enter the number instead of (2004)
L117 enter the date and year of the decision
L149 (total 48 birds)? After replicate
L152, 157, 158, 164, etc. 4 °C, -80 °C , space between number and °C is required according to the instructions
L247 „small intestine” instead of jejunal
L263 Bray-Curtis
L272 Lactobacillus – none “s”
L277 HJ-3 instead of HJ-2
L287 [16,17] , Liu et al. [18]
L333 [11,28,29]
L373 Data curation: then Initials of surname and first name, For other names of the procedure then initials instead of first and last names (see other articles in Animals)
L389+ Section must be performed according to instructions for authors
Reviewer 4 Report
Comments and Suggestions for Authors
The Addition of Hot Water Extract of Juncao-Substrate Ganoderma Lucidum Residue to Diets Enhances Growth Performance, Immune Function, and Intestinal Health in Broilers
Dear Authors,
The manuscript interesting and quite well prepared. Extract used in experiment have positive effect for broiler chickens and can be used in maximal dose planed in research. Some corrections are required in the text of manuscript. Light regimen during experiment must be describe more precisely. Nutrients level in diets is calculated, but information about chemical composition of main ingredients can be added in new table. Bacteria genus must be describe more precise. More references must be added to the Introduction and Reference subsection must be adapted to standards.
Below I add some suggestions helpful in this process:
Line 3
In consequence of title of manuscript, words ‘Diets Enhances’ should start from capital letters.
Line 29-277
In text of is used ‘P-value, must be used p-value.
Instead of P = 0.033, p = 0.033 must be used.
Footer, lower left corner of the page
Information about Journal, year of publication and volume must be changed from year 2023 to year 2024 and actual volume
Lines 54-60
Two, three references must be added to this part of Introduction.
Lines 65-67 and 68-69
Reference to each sentence must be also added.
Lines 74, 81 and 84
In case of reference in text of manuscript, year of publication is not required, ie. in line 74: Song et al. [8] is enough, the same with Chen et al. [10] and Khan et al. [11].
In this case numeration on the end of sentence is not required.
Lines 87-89
References required.
Lines 110-111
Please check information or give more details about light regime, because in text is given only: under 23 hours. In this case conditions in majority of rearing programs, light regime is changed from 23 hours in first one-two days to 20 hours of light to 10th day of life of birds, and after to 16 hours from 10th day It is done to increase welfare of broiler chickens in order to even out the herd, increase feed conversion, reduce energy consumption and reduce the number of mechanical injuries and limb problems in birds.
Line 118
In line 125 is information about that nutrient levels were calculated, but they are similar in case of four treatments. Maybe it is possible to add information in new table about those 6 nutrients (ME, CP, Ca, non-phytate P, Lys and Met+Cys) for corn, soybean meal and expanded soybean?
Number of decimals in table must be equal in rows (zero must be added).
Lines 207-210
Analysis was conducted used one-way ANOVA, but in this case information about normal distribution of data in groups/treatments is required (Shapiro-Wilk’s test). Homogenity of variance in treatments is also important in treatments, after using Levene’s test or other.
Line 217
p-value must be used in header of Table 5.
Line 219
In text of manuscript under Table 5 is part of sentence ‘…without the same superscript differ (P < 0.05).’ , better is to use: ‘…without the same superscript significant differ at p< 0.05.’
Line 227
p-value must be used in header of Table 6.
Line 230
In Figure 1 description second and third sentence can be rearranged. Ie.: Values are presented as a mean ± sd. Letters in case of each cytokine describes significant differences between groups/treatments at p < 0.05.
Line 251
Information about significance level must be added in description of Figure 2.
Letters in case of each cytokine or jejunal tight-junction protein describes significant differences between groups/treatments at p < 0.05.
Line 252 and 253
p-value must be used in header of Table 7 and 8.
Line 279
Figure 3E
Genus of bacteria is specified in legend, but in this case: Romboutsia, Turicibacter, Faecalibacterium, Blautia and Lactobacillius specification sp. or spp. must be added to genus name.
Line 283
p-value must be used in header of Table 9.
Description of significance level must be added under the table as in lines 218-219.
Line 284
p-value must be used in header of Table 10.
Description of significance level must be added under the table as in lines 218-219.
Genus of bacteria is specified in first column, but in this case: Romboutsia, Turicibacter, Faecalibacterium, Blautia and Lactobacillius specification sp. or spp. must be added to genus name.
Desulfobacterota, control treatment, c letter must be used in superscript form.
The same with Romboutsia (sp. or spp.), HJ-3 treatment, b letter must be used in superscript form.
Line 287
Liu et al. [18], without the year of publication.
Lines 354-362
Blautia and Turicibacter genus must be more precise specified (sp. or spp.).
Lines 389-476
References subsection must be adapted to Animals/MDPI standards mentioned in Instructions for Authors, in order:
Authors, Title, Journal’s name (abbreviated from with dots/points), year of publication, volume, pages, Doi link
Ie., no 1.:
Qin, X.; Fang, Z.; Zhang, J.; Zhao, W.; Zheng, N.; Wang, X. Regulatory effect of Ganoderma Lucidum and its active components on gut flora in diseases. Front. Microbiol. 2024, 15, 1362479. https://doi.org/10.3389/fmicb.2024.1362479
Author Response
请参阅附件

Round 2
Reviewer 3 Report
Comments and Suggestions for Authors
The purpose of this experiment was to investigate the effects of Hot Water Extract of Juncao-substrate Ganoderma Lucidum Residue (HWE-JGLR) on the immune function and intestinal health of yellow-feather broilers. The results obtained are important for poultry practice.
General comments:
In my opinion, the article should be supplemented with the following information:
No information about the type of scale for recording BW, FI and measurement accuracy
The References section must be done according to the instructions for the authors:
The volume number must be in talic
Name initials instead of full name are required for positions 2-10, 18, 19, 21, 22, 24-27, 34-36, 39, 40
For page ranges use long dash () from the symbol function, instead of short (-) from the keyboard
We do not use the word "and" before the last author of the article
Specific comments
Affiliation designations as superscript 1, 2 are required, a superscript asterisk for the corresponding author, at the end Contributed without designations
L85, 307, 386 should be „Lactobacillus” instead of Lactobacillu, add „s”
Tables 1, 3, 6 from the beginning of the next page
In Table 5, IBW instead of IW; in the explanations IBW – initial body weight
L272 (p > 0.05), space before "0.05"
L285, 287 space after "p"
L392, 396 "spp." instead of spp - add "a dot" after spp
L454 please separate concatenated names
L480 Seweryn, E; Zioła, A; Gamian, A. instead of current form

Reviewer 4 Report
Comments and Suggestions for Authors
The Addition of Hot Water Extract of Juncao-substrate Ganoderma lucidum Residue to Diets Enhances Growth Performance, Immune Function, and Intestinal Health in Broilers
Thank you for revision process, all suggestions were included in new version of manuscript.
This time only several corrections mainly from point of view edition of text:
Line 2
Addition word in title of manuscript in this form must starts from capital letter.
Line 3
Ganoderma lucidum, in title of manuscript in case of binominal nomenclature italics must be used (genus name from Capital letter, species name from small letter).
Lines 13 and 25
The same like in line 3 (genus name from Capital letter, species name from small letter).
Line 40
Must be spp. (with dot/point on the end, without italics).
In text of manuscript is Romboutsia app, must be Romboutsia spp.
Lines 305-310
spp. (with dot/point on the end, without italics).
Line 307
Lactobacillus spp. (lack of s on the end in manuscript).
Line 386
Lactobacillus
Lines 430-537
All abbreviations of Journal Names must end with dot/point.
Ie. Front. Microbiol., Front. Pharmacol. and so on...
On the end of each reference doi number/link must be also added, ie. in case of reference 1: https://doi.org/10.3389/fmicb.2024.1362479
Round 3
Reviewer 3 Report
Comments and Suggestions for Authors
The purpose of this experiment was to investigate the effects of Hot Water Extract of Juncao-substrate Ganoderma Lucidum Residue (HWE-JGLR) on the immune function and intestinal health of yellow-feather broilers. The results obtained are important for poultry practice.
General comments:
In my opinion, the article should be supplemented with the following information:
No information about the type of scale for recording BW, FI and measurement accuracy
Detailed comments:
L105+ 2.1.; 2.2.; 2.3.; 2.4., etc. requires ‘a dot’ after the second number.
L153: 2-3 °C instead of current form
L308-313: spp. instead of spp – please add ‘a dot’ after spp
